# High-speed and on-chip graphene blackbody emitters for optical communications by remote heat transfer

Yusuke Miyoshi[1], Yusuke Fukazawa[1], Yuya Amasaka[1], Robin Reckmann[1,2], Tomoya Yokoi[1], Kazuki Ishida[1], Kenji Kawahara[3], Hiroki Ago[3] & Hideyuki Maki[1,4]

High-speed light emitters integrated on silicon chips can enable novel architectures for silicon-based optoelectronics, such as on-chip optical interconnects, and silicon photonics. However, conventional light sources based on compound semiconductors face major challenges for their integration with a silicon-based platform because of their difficulty of direct growth on a silicon substrate. Here we report ultra-high-speed (100-ps response time), highly integrated graphene-based on-silicon-chip blackbody emitters in the near-infrared region including telecommunication wavelength. Their emission responses are strongly affected by the graphene contact with the substrate depending on the number of graphene layers. The ultra-high-speed emission can be understood by remote quantum thermal transport via surface polar phonons of the substrates. We demonstrated real-time optical communications, integrated two-dimensional array emitters, capped emitters operable in air, and the direct coupling of optical fibers to the emitters. These emitters can open new routes to on-Si-chip, small footprint, and high-speed emitters for highly integrated optoelectronics and silicon photonics.

[1] Department of Applied Physics and Physico-Informatics, Keio University, Yokohama 223-8522, Japan. [2] Faculty of Electrical Engineering and Information Technology, RWTH Aachen University, 52074 Aachen, Germany. [3] Global Innovation Center (GIC), Kyushu University, Fukuoka 816-8580, Japan. [4] PRESTO JST 4-1-8 Honcho, Kawaguchi, Saitama 332-0012, Japan. These authors contributed equally: Yusuke Miyoshi, Yusuke Fukazawa.  Correspondence and requests for materials should be addressed to H.M. (email: maki@appi.keio.ac.jp)

Nanocarbon-based optoelectronic devices are promising candidates for the high-speed, uncooled, and on-chip optical communication devices, such as light sources[1–4], photodetectors[5], and modulators[6]. Recently, carbon nanotube (CNT)-based light emitters, where CNTs are directly formed on silicon substrates, have been reported as light sources, based on electroluminescence by electron-hole recombination[1,7,8] and blackbody radiation by Joule heating[9–11]. It is a great feature that these devices work as on-chip and small-footprint light emitters in the near-infrared (NIR) region, including tele-communication wavelength[10,11]. In addition, the high-speed light emission with a response time of ~100 ps has been recently demonstrated for CNT-based blackbody emitters, where the high-speed emission is explained by the extremely fast temperature response of CNT films, due to unique thermal properties of its one-dimensional structure[11,12]. Recently, optical communications are also demonstrated by using CNT emitters on silicon chips[4,11].

Graphene, which is another nanocarbon material, has unique two-dimensional properties in electronic[13], optical[14,15], and thermal properties[16], which have been applied for optoelectronic devices. Graphene-based blackbody emitters (gray-body emitters for thin graphene due to the low emissivity of 2.3 % per layer[17–22]) are also promising light emitters on silicon chip in NIR and mid-infrared region, just like CNT-based blackbody emitters[17–27]. In addition, the significant advantage of the emitter with graphene instead of CNTs is the planar, uniform, and bright light emission due to its two-dimensional geo-metry[19,20]. However, although graphene-based blackbody emit-ters have been demonstrated under steady-state conditions or relatively slow modulation (100 kHz) in large area devices[26], the transient properties of these emitters under high-speed modula-tion have not been reported to date. Also, the optical communications with graphene-based emitters have never been demonstrated.

Here, we report a highly integrated, high-speed, and on-chip blackbody emitter based on graphene in NIR region including telecommunication wavelength. Under a rectangular voltage, a fast response time of ~100 ps, corresponding to ~10 GHz mod-ulation, has been experimentally demonstrated for single and few-layer graphene, and the emission response properties are strongly affected by the degree of graphene contact with the substrate, which depends on the number of graphene layers. The mechanisms of the fast modulation in this emitter are elucidated by performing theoretical calculations of the heat conduction equations considering the thermal model of emitters including graphene and a substrate. The simulated results indicate that the fast response properties can be understood not only by the classical thermal transport of in-plane heat conduction in gra-phene (dominated by hot spots) and heat dissipation to the substrate, but also by the remote quantum thermal transport via the surface polar phonons (SPoPhs) of the substrates[28–34]. Moreover, we experimentally demonstrate optical communica-tions at 1 and 50 Mbps based on eye-pattern analysis and real-time waveform detection by using multi- and few-layer graphene, respectively, integrated two-dimensional array emitters with a chemical vapor deposition (CVD) grown graphene, capped emitters operable in air, and the direct coupling of optical fibers to the emitters.

## Results

**Device structure and emission properties**. A schematic picture and a typical optical-microscope image of the fabricated device are shown in Fig. 1a, b, respectively. The electrodes were designed to have a 50 Ω characteristic impedance using coplanar

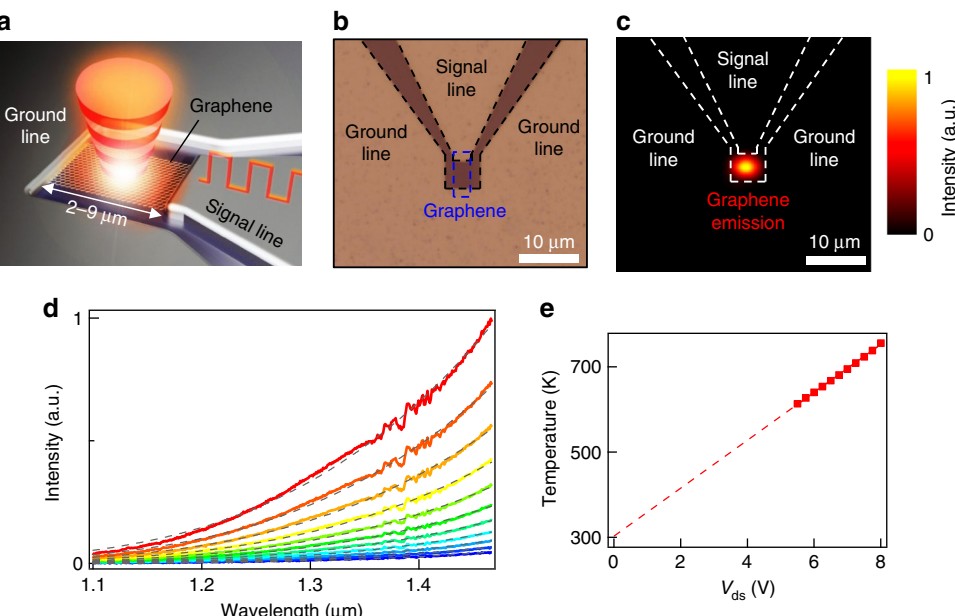

**Fig. 1** Device structure and emission from a graphene device under DC bias. **a** A schematic illustration of graphene blackbody emitter. Square graphene sheet on a SiO₂/Si substrate is connected to source and drain electrodes (signal and ground lines). Modulated blackbody emission is obtained from graphene by applying input signal. **b** An optical-microscope image of the graphene emitter with a coplanar waveguide. Blue and black dashed lines indicate square graphene sheet and the coplanar electrodes, respectively. **c** IR camera image of the thermal emission at $V_{ds} = 4$ V for the emitter shown in (**b**). The bright emission is localized to a small hotspot in graphene. **d** Emission spectrum from a single-layer graphene emitter for $V_{ds} = 5.5–8$ V with steps of 0.25 V. The small peaks located at wavelengths around 1.4 μm are due to the light absorption by water in air and remain even after spectral correction (see Methods). The gray broken curves are fitting results based on Planck's law. **e** Graphene temperatures obtained by the Planck's-law fittings in (**d**). Their temperatures depend linearly on the applied voltages

transmission lines for high-speed operation under high-frequency bias voltage[11]. Graphene was formed at the termination of the transmission lines. In this study, few-layer and multi-layer (≳5) graphene has been prepared by the mechanical exfoliation method with a scotch tape, and single-layer graphene has been prepared by CVD growth[35]. DC bias voltage dependences of the current for these emitters are shown in Supplementary Note 3. Under continuous DC bias voltage, the bright emission from the graphene between two electrodes is observed with a NIR camera (Fig. 1c). The emission from these devices has a broad spectrum in NIR region including telecommunication wavelength, and the emission intensity increases with increasing applied voltages (Fig. 1d). Because these spectra are fitted to Planck's law (broken lines in Fig. 1d), the observed emission from graphene represents blackbody radiation generated by Joule heating[10,11,17,19–22]. The graphene temperatures, which are obtained from these fittings, depends linearly on the applied voltage due to electron scattering with optical phonons caused by Joule heating[22,36,37] and can reach about 750 K at $V_{ds}$ = 8 V (Fig. 1e).

**Time-resolved emission measurements**. The response speeds of light emitters were experimentally elucidated by the time-resolved emission measurement based on a single-photon counting method with a Geiger-mode avalanche photodiode (APD) under a rectangular bias voltage[11]. Figure 2a, b show the time-resolved emission intensities for a single-layer-graphene device as a function of width and amplitude of rectangular inputs, respectively (see Supplementary Note 4 for the results of the emitter with three-layer graphene). The emission intensity quickly responds to the applied rectangular input, and the response time defined by the time of 10–90% intensity is ~0.4 ns, which is dominated by the relatively slow response time of a signal generator (~0.5 ns), i.e., the intrinsic response time of this device is faster than the experimentally obtained one. Since the emission intensity of blackbody radiation is roughly proportional to $T^4$,

taking into account the Stefan–Boltzmann law, only the high-voltage region of the input signal, i.e., high-temperature region due to Joule heating in graphene, contribute to the behavior of the time-resolved emission. For faster operation, we also demonstrated the short-pulsed light generation with a high-speed pulse generator as shown in Fig. 2c. Very-short-pulsed emission with a rise time and width of ~100 ps and ~200 ps, respectively, can be generated by applying a pulsed voltage of 100 ps in width.

In these devices, the emission responses consist of two components: an initial fast response with a rise time of ~100 ps and a second slow response of the temperature rise. Figure 2d shows the time-resolved emissions for the emitters with single, three, eight, and twenty-eight-layers graphene. The intensity ratios of the fast and slow components roughly depend on the number of graphene layers. For the device with single-layer graphene, the initial fast response can be clearly seen and the subsequent intensity is almost flat. In contrast, for multi-layer graphene (8 and 28 layers in Fig. 2d), the component of the initial fast response is small, and the second slow component mainly dominates the overall intensity response. The few-layer graphene (three layers in Fig. 2d) shows intermediate properties of the emission responses, where dominant component of initial or second response changes from sample-to-sample. These results roughly indicate that the second slow component is emphasized compared to the initial fast component with increasing the number of graphene layers.

**Theoretical calculations and mechanisms of high-speed emission**. Since the temperature response of graphene dominates the response of blackbody emission, we theoretically elucidate the transient temperature of graphene under bias voltage to investigate the mechanisms of the observed high-speed emission. Here, we numerically calculated the transient temperature distribution of graphene and $SiO_2$ substrate under bias voltages by solving the simultaneous heat conduction equations of one-dimensional

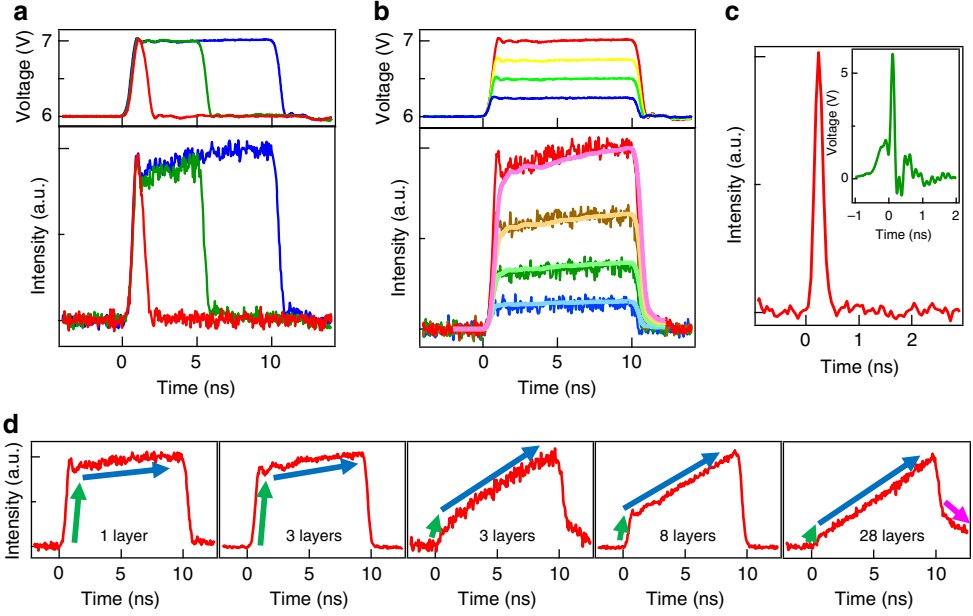

**Fig. 2** Experimental high-speed light emissions. Time-resolved emission under a rectangular bias voltage with **a** different pulse widths (1, 5, and 10 ns in width and 6 V–7 V in height) and **b** different pulse amplitudes (10 ns in width and 6 V–6.25, 6.5, 6.75, and 7 V in height) to single-layer graphene device. **c** Very-short-pulsed light generation with a width of 200 ps for three-layer graphene device (red curve) by applying a pulsed input voltage with a width of 100 ps and amplitude of 0–6 V (green curve). **d** Time-resolved emission for the emitters with single, three, eight, and 28-layers graphene. The emission responses consist of two components: an initial fast response with a rise time of ~100 ps (green arrows) and a second slow response of the temperature rise (blue arrows), whose intensity ratios roughly depend on the number of graphene layers. For thick graphene, the slow fall component of the emission is observed as shown by pink arrow

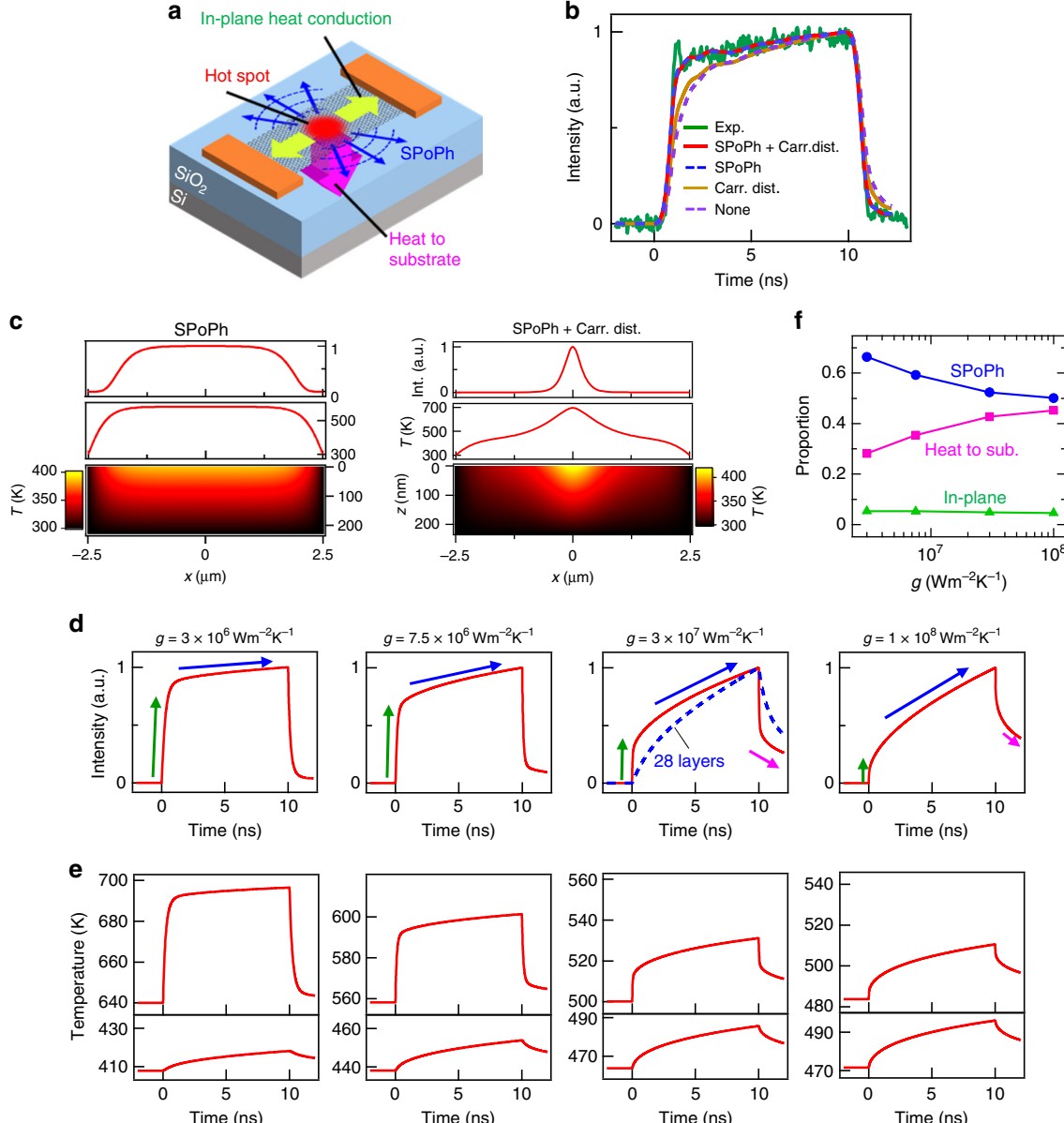

**Fig. 3** Theoretical calculations and mechanisms of high-speed emission. **a** Schematic picture of the thermal transport model in graphene device: heat dissipation to substrate, in-plane heat conduction, enhanced by a hot spot, and quantum remote heat transfer by SPoPh. **b** Numerically calculated time-resolved emission intensities for the model with/without the consideration of the spatial carrier distribution (with: red and brown, without: blue and purple broken curves) and the remote SPoPh scattering(with: red and blue, without: brown and purple curves) in comparison with the experimental result (green curve) in Fig. 2a under 10-ns-width rectangular input. **c** Top and middle: emission intensity and temperature distribution of graphene, respectively. Bottom: two-dimensional temperature distribution of a $SiO_2$ substrate. Left and right results are for the model with and without the consideration of spatial carrier distribution, respectively. In this calculation, the graphene length and the $SiO_2$ thickness are 5 µm and 230 nm, respectively. For the result with carrier distribution, a hot spot is generated at the Dirac point. **d** Emission intensity response simulated for the single-layer graphene emitter as a function of the thermal conductance $g$ between graphene under ideal rectangular voltage (width: 10 ns, amplitude: 6–7 V). The emission intensity is calculated by the integral of Planck's law over the wavelength region from 0.9 to 1.6 µm. Broken blue curve is the simulated result taking into account the number of layers (28 layers). Green, blue, and pink arrows correspond to the arrows in Fig. 2d. **e** Transient temperature of the center of graphene (top) and the uppermost $SiO_2$ layer beneath the graphene (bottom) as a function of $g$ under the same condition of (**d**). **f** Proportion of heat dissipation mechanisms (SPoPh, heat to a substrate and in-plane heat conduction) as a function of $g$

graphene and two-dimensional $SiO_2$ substrate (see Supplementary Note 2). In the simple and classical heat conduction model for the graphene device under DC current heating (i.e., steady state), the temperature distribution of graphene can be understood by the thermal transport of heat dissipation to the substrate, which is determined by the physical contact between graphene and substrate[16,38–42] and in-plane heat conduction, which is strongly affected by the hotspot generation near the Dirac point

due to carrier-density distribution (Fig. 3a)[16–18,23,24]. Recently, it has been found that the quantum remote phonon scattering by SPoPhs in the $SiO_2$ polar substrate plays an important role in electron[28–32,36,37] and thermal[31–34,36,37] transport in nanocarbon-based devices. Although it has been pointed out that the SPoPh scattering can strongly affect the steady-state graphene temperature under DC Joule heating[31–33,36,37], it is an open question how the SPoPhs affect the transient temperatures of

graphene devices under high-speed modulation of Joule heating. Here, we theoretically investigate the transient temperature $T$ distribution of graphene devices under rectangular voltage input by performing numerical calculations of a two-dimensional heat conduction equation taking into account the classical thermal transport in graphene and a substrate, the hot spot generated by the spatial carrier-density distribution and the direct heat dissipation via SPoPhs (Fig. 3a)

Figure 3b shows the numerically calculated time-resolved emission intensities for the model with/without the consideration of the spatial carrier distribution and the remote SPoPh scattering in comparison with the experimental result in Fig. 2a under 10-ns-width rectangular input. For the models with the spatial carrier distribution, a hot spot is formed at the Dirac (neutrality) point position, where the carrier density is minimized (Fig. 3c)[16–18,23,24]. Since the rate of temperature change is proportional to the spatial curvature of the graphene temperature given by $\kappa_{gr}(\partial^2 T/\partial x^2)$, where $\kappa_{gr}$ is the thermal conductivity of graphene, the formation of a hot spot can contribute to the faster temperature response (i.e., the faster emission response) of graphene emitters. However, the existence of a hotspot by itself is insufficient to explain the fast response of the experimentally observed emission (Fig. 3b). Remote SPoPh scattering can also contribute to the faster temperature response of graphene emitters because the surface phonon polaritons excited by the remote SPoPh scattering of graphene carriers can propagate along the substrate surface over a long distance of $\gg 10\,\mu m$[43–45]. In this case, the thermal energy can be directly dissipated towards the outside of the device system without a temperature rise of the SiO$_2$ substrate just beneath the graphene. With the model taking into account the SPoPh scattering and the hot spot, the simulated emission responses can be fit very well to the experimental emissions in Figs 2b and 3b, which exhibit the clear initial response of emission intensity and the almost flat response of the subsequent region due to suppression of temperature rise in the substrate by the SPoPh dissipation.

To understand the detailed mechanisms of the emission responses in Fig. 3b, we calculated the transient temperature response of graphene and a substrate under the ideal rectangular bias voltage. Figure 3d, e show the emission intensity from the graphene emitters and the transient temperature of the center of graphene and the uppermost SiO$_2$ layer beneath the graphene as a function of the thermal conductance $g$ between graphene and the substrate, respectively. The simulated results show the two components of temperature response as well as the experimental results in Fig. 2. In the initial region with fast response, only the graphene temperature quickly rises by Joule heating, independently of the substrate temperature. In the subsequent second region with slow response, the temperature response of graphene correlates with the uppermost SiO$_2$ temperature, indicating that the graphene temperature rises according to the slow temperature rise of the substrate beneath the graphene. The temperature response in both regions are strongly affected by the thermal conductance $g$ between graphene and the substrate[16,36–42]. For a high $g$, in which the emission behaviors are less affected by the other heat transports of the SPoPh and the in-plane heat conduction of graphene (i.e., $\kappa(\partial^2 T/\partial x^2)$), the response speed of the initial region becomes high because the temperature relaxation time $\tau$ is simply given by $\rho C/g$[11,46,47]. However, for a high $g$, the degree of the temperature rise in the initial fast region is suppressed by high heat dissipation to the substrate and its degree in the second slow region is emphasized by high thermal coupling between graphene and the substrate; consequently, the overall emission response is dominated by the second region with a slow temperature response. In contrast, with decreasing $g$, the dominant mechanism of the temperature response in the initial

fast region crosses over from heat dissipation toward substrate ($g(T-T_0)$) to the SPoPh and in-plane heat conduction of graphene ($\kappa(\partial^2 T/\partial x^2)$). The degree of the temperature rise in the initial fast region increases and the influence of the substrate temperature in the second slow region is suppressed; that is, the shape of the overall temperature response approaches the shape of the rectangular input voltage. In this condition, the temperature response time of the initial fast region is in the order of 100 ps. We note that the experimental results (i.e., rectangular and fast emission response in Fig. 2) could not be explained by the simulated heat conduction without the SPoPh transport (e.g., Fig. 3b). As shown in Fig. 3c, the spatial carrier distribution in a graphene emitter can cause a hot spot located at the Dirac point, as also shown in previous reports[17,18,23–25]. Although a hot spot can increase the in-plane heat conduction due to increase of $\kappa$($\partial^2 T/\partial x^2$) in principle, the contribution of the hot spot generation to the fast temperature response is not that high from the numerical calculation as shown in Fig. 3b, f. However, SPoPh can strongly contribute to the fast and rectangular emission behaviors in the low-$g$ condition, because both the increase of the speed of the initial temperature rise and the suppression of the subsequent slow region can be realized by SPoPh remote thermal transport due to the long propagation length of surface phonon polaritons and the suppression of the temperature rise of the substrate beneath the graphene.

These simulated results can explain the experimental emission response depending on the number of layers. As shown in Fig. 2d, the emitters with single- or few-layer graphene on SiO$_2$ exhibit a clear rectangular emission response under a rectangular input. This quick rectangular response can be qualitatively understood by the temperature response for the emitter with low thermal conductance $g$ (Fig. 3d), because the ripples of single- or few-layer graphene on SiO$_2$ substrate cause poor thermal contact between graphene and substrate[48–50]. In contrast, the emitters with multi-layer graphene exhibit the distorted slow emission response under a rectangular input (Fig. 2d) because a high $g$ due to the atomically-flat surface of multi-layer graphene leads to the slow temperature response, as shown in Fig. 3d. We note that not only $g$, but also $\kappa$ of multi-layer graphene is higher than that of single-layer graphene[16]; however, the distorted emission response of multi-layer graphene in Fig. 2d cannot be explained by the model without high $g$. From the comparison of the experimental and simulated emission response, we determine the $g$ values to be ~$3 \times 10^6$ for single- and few-layer graphene and ~$3 \times 10^7\,\mathrm{Wm}^{-2}\,\mathrm{K}^{-1}$ for multi-layer graphene. Although the trend that $g$ is low for single- and few-layer graphene is consistent with the previous report[40], these obtained values are relatively low compared with the previously reported $g$ ranging from ~$10^7$ to $10^8\,\mathrm{Wm}^{-2}\,\mathrm{K}^{-1}$ for single- and multi-layer graphene[16,23,36,38–42]. The $g$ value can be estimated to be larger when the remote thermal transport via SPoPhs is included in the thermal conductance $g$ as the previously reported steady-state heat conduction and the effect of the direct heat dissipation via SPoPhs does not clearly appear for larger graphene than the SPoPh propagation length (i.e., $\gg 10\,\mu m$)[43–45]. Our results imply that the direct measurement of the transient temperature is an effective method to precisely elucidate the in-plane and out-of-plane thermal transport in the graphene devices, and the time-resolved emission measurement, demonstrated in this study, is one of the effective method for the transient temperature investigation.

**Optical communications.** To investigate the practical applications of these graphene emitters for optical communications, we demonstrate the real-time detection of the modulated light from the graphene emitters by using a conventional photoreceiver with

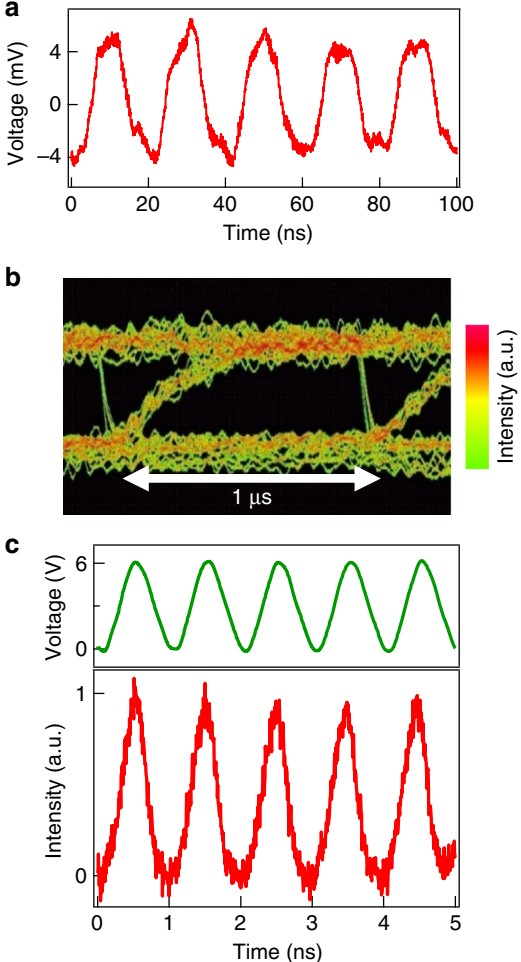

**Fig. 4** Real-time optical communications and high-speed continuous modulation. **a** Real-time detection of the emission from three-layer graphene device at 50 MHz achieved by using a normal-mode APD photoreceiver under a rectangular voltage (0 V–8 V in height). **b** Eye-diagram of emission from 85 layer graphene device at data rate of 1 Mbps under random square voltage of 0 V–4.3 V. **c** High-speed emission modulation (red curve) from three-layer graphene device under a continuous input (0 V–6 V in height) of 1 Gbps (green curve) by time-resolved emission measurements

(Fig. 4c). High-speed modulation synchronized with the input signal of 1 Gbps is experimentally demonstrated. Ideally, the communication speed of ~10 Gbps is possible taking into account the ~100-ps rise time of the graphene emitters in Fig. 2.

**Highly integrated graphene emitters**. As mentioned above, the high-speed blackbody emitters can be fabricated by using CVD-grown graphene as well as exfoliated graphene. Since the CVD-grown single-layer graphene can be uniformly transferred with the large size of >1 cm$^2$ on the silicon wafer, it is a great advantage that many light emitters can be directly integrated at any positons on silicon chip[20,35]. We fabricated two-dimensional array of graphene emitters, which are high dimension and density by using unsuspended graphene compared with earlier work[20]. Figure 5a shows a fabricated device with a $2 \times 8$ array of light emitters with CVD graphene, which has a size of $2 \times 2$ μm and a pitch of 3 μm. From this device, the arrayed light emissions with relatively uniform intensity can be obtained from all graphene channels as shown in Fig. 5b. This indicates that the graphene-based emitter is a promising candidates for high-density emitters on silicon chips because the array density of fabricated graphene emitters is hundreds of times higher than typical semiconductor array light emitters[52].

Although all graphene emitters described above are measured in a high vacuum to avoid the damage of graphene by reaction with oxygen in air, we fabricated the graphene emitters encapsulated by $Al_2O_3$ insulator. Such encapsulation technique was previously demonstrated by using hexagonal boron nitride (h-BN), and this technique allowed the operation of large area devices for more than 1000 h in air[21]. We grew the $Al_2O_3$ capping layer by an atomic layer deposition (ALD) method, which is used in a conventional silicon-based semiconductor process. The conductance of the device was decreased to ~50 % by the deposition of $Al_2O_3$ due to charged impurity scattering and SPoPh scattering by the capping layer[53]. As shown in Fig. 5c, the $Al_2O_3$-capped emitters can operate in air for more than 100 h. Furthermore, by using this $Al_2O_3$-capped emitters, we also demonstrated the direct coupling of this capped light emitter to a multimode optical fiber in air owing to their small footprint and planar device structure (Fig. 5d). High-speed light emission could be directly observed though this optical setup (Fig. 5e). This indicates that the light emitters on silicon chips can be directly coupled to optical fibers in small footprint without optical elements.

## Discussion

We have demonstrated graphene-based blackbody emitters, which have a high modulation speed (100-ps response time) and small footprint (~1 μm$^2$) at NIR wavelength including tele-communication wavelength. The 100-ps response time of this graphene emitter is faster than that of the conventional semiconductor light emitting diodes (~MHz) and is comparable to that of the laser diodes (~GHz). In addition, we have reported the application of graphene-based emitters for optical communications via optical fiber with the real-time signal transmission (1 and 50 Mbps) by using multi- and few-layer graphene, respectively. The high-speed modulation can be understood by the fast thermal relaxation dominated not only by the classical thermal transport of the in-plane and out-of-plane heat conduction, but also by the remote quantum thermal transport via the SPoPhs of the substrates. These findings suggest that the emission intensity and modulation speed of a graphene emitter can be controlled by the number of layers of graphene and the physical contact between graphene and substrate, indicating that the emission properties can be designed by the device structure. Since SPoPhs

a normal-mode APD[11]. Figure 4a shows the real-time detection of the modulated emission from three-layer graphene under 50-MHz rectangular input. The detection waveform synchronized with the graphene emission is clearly obtained with an oscilloscope. Note that the slow response of the obtained waveform is ascribed to the narrow bandwidth (50 MHz) of the function generator for the graphene emission and the amplifier for detection. In addition, we also demonstrated the optical communication investigated by an eye-pattern analysis at 1 Mbps (Fig. 4b). In this measurement, multi-layer graphene (~85 layers) was used because of higher emission intensity and higher durability of the graphene devices under emission with increasing the number of graphene layers[26,51]. A completely open eye is observed indicating that the optical commutation is possible at 1 Mbps. The relatively slow and asymmetric response observed in the eye pattern corresponds to the slow emission response in multi-layer graphene as shown in Figs 2d and 3d (see Supplementary Note 5). Furthermore, we investigate the ability of this emitter to high-speed optical communications by time-resolved emission measurements under a continuous-wave input voltage

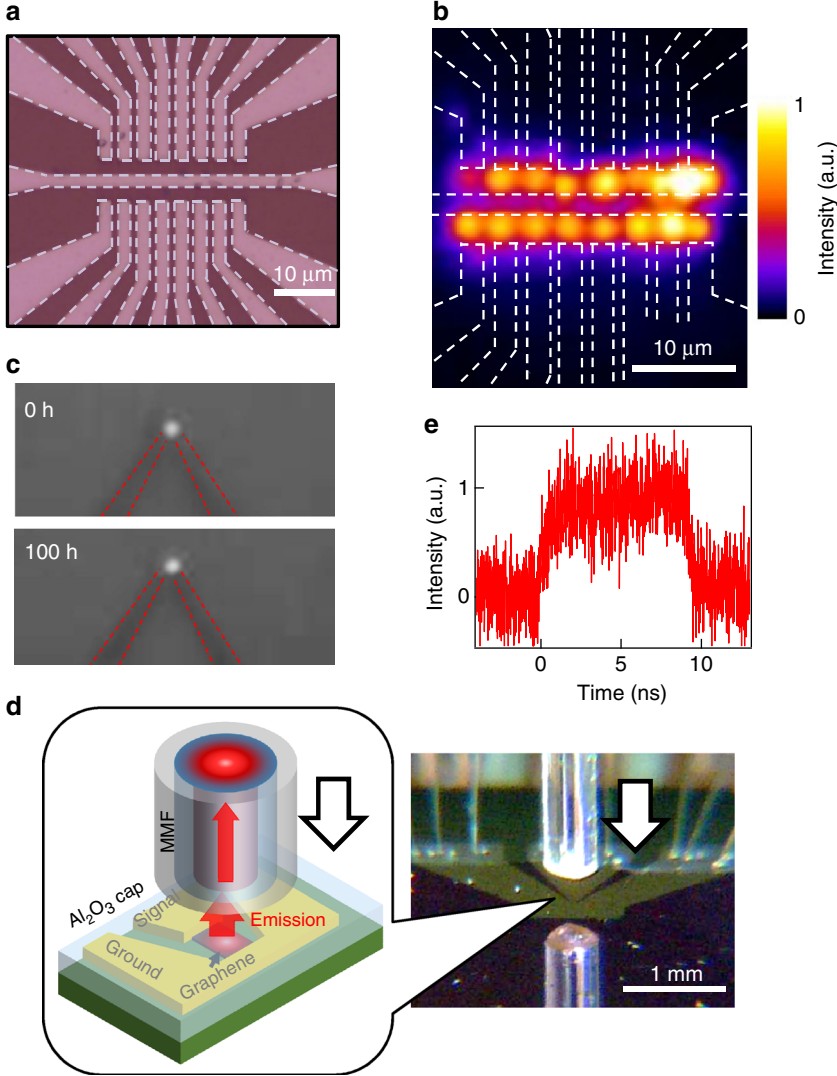

**Fig. 5** Highly integrated graphene emitters. **a** An optical image of the fabricated device of the 2 × 8 array emitters, whose size and pitch are 2 × 2 μm and 3 μm, respectively. Dashed lines indicate Pd electrodes. **b** NIR camera image of the emission for the array device at $V_{ds} = 10$ V. All devices are connected to the same voltage source, but an adjustable series resistor is introduced for every emitter to equalize the emission intensity of every emitter. **c** NIR camera images of the light emission from $Al_2O_3$-capped emitters with three-layer graphene after 0 and 100 h of continuous work in air. No significant change is observed after 100 h. **d** Optical image of the direct coupling of the capped light emitter to the multimode optical fiber (MMF) in air. In this picture, MMF is approaching the emitters. Inset: schematic image of the direct coupling of a capped light emitter to a MMF. **e** High-speed light emission measured by the direct coupling shown in (**d**)

can be excited in the polar substrate, such as $HfO_2$, $Al_2O_3$ and h-BN[33,54], the effect of SPoPh can be modified by changing the substrate. In particular, two-dimensional materials such as h-BN have recently attracted attention owing to their unique properties of their flat surfaces[49,55], anisotropic structure, and two-dimensional polariton transport[56], and these unique properties are applicable to the graphene light emitters[21,26]. Graphene light emitters are greatly advantageous over conventional compound semiconductor emitters because they can be highly integrated on silicon chip with simple lithography processes by using large scale CVD graphene. In addition, it has been reported that the optoelectronic devices with nanocarbon materials are compatible with integrated silicon photonics and three-dimensional integrated circuits[2,4,5,8]. Hence, graphene-based emitters can be directly combined with the silicon platform with three-dimensional integrated electronics and integrated silicon photonics, and they can open new routes to highly integrated optoelectronics, e.g., optical interconnects.

## Methods

**Device fabrication**. We fabricated graphene-based light emitters on a thermally oxidized (230 nm) silicon substrate (Fig. 1a, b). We used two kinds of graphene: (i) mechanically exfoliated few-layer or multi-layer graphene with scotch tape and (ii) CVD-grown single-layer graphene transferred onto a $SiO_2$/Si substrate using poly(methyl methacrylate) (PMMA)[35]. The number of graphene layers are derived from the intensity ratio of 2D and G peaks in Raman spectroscopy for single- and few-layer graphene and from the thickness measured by an atomic force microscope for multi-layer graphene. The graphene on the substrate was patterned by e-beam lithography and $O_2$ plasma etching with a Ni mask, which is subsequently removed by hydrochloric acid solution. Ti/Pd electrodes were deposited as source and drain electrodes on the graphene. The graphene sizes of device channels were 5 × 5 μm for the device with single-layer graphene, 6 × 9 μm for the device in Fig. 4b, 2 × 2 μm for the array device in Fig. 5a, b, and 3 × 4.5 μm for the other devices. These electrodes were designed to be coplanar waveguide with a characteristic impedance of 50 Ω on an undoped silicon substrate for high-speed light emitters (except for the array device in Fig. 5a, b)[11]. A gate bias cannot be applied because of the use of an undoped substrate. The capped emitters for the air operation (shown in Fig. 5c) are fabricated by deposition of 75-nm-$Al_2O_3$ insulator on the fabricated graphene emitters by an ALD method.

**Emission measurements**. The emission from the graphene devices were measured by a micro-photoluminescence measurement system, where the emitted light was collected through a quartz optical window of a high-vacuum sample chamber and a microscope objective lens at room temperature, as reported in previous reports[7,10,11]. The $Al_2O_3$-capped emitters were measured in air. Two-dimensional emission images and emission spectra of the emission was detected by an InGaAs CCD camera and an InGaAs linear detector with a spectrometer, which has the wavelength range from 0.9 to 1.6 µm, respectively. The relative spectral response of the measurement system including the optical path and the detector was measured with a standard light from a blackbody furnace, and all spectra were corrected accordingly. Bias signals were applied through a signal line of the coplanar device using a DC voltage source, a signal generator, a function generator and a pulse generator for steady-state emission, time-resolved high-speed emission measurement, real-time optical communications, or pulsed light generation, respectively. In the time-resolved emission measurements based on a single-photon counting method, the emitted light was guided to the Geiger-mode InGaAs APD (wavelength range: 0.9–1.7 µm) through a fiber coupler and a multimode graded-index optical fiber, and the time-resolved emission intensity was obtained using a time-correlated single-photon counting module (see Supplementary Note 1)[11]. The time-resolved measurements based on a single-photon counting method is effective method to investigate the response speed of the low-intensity light sources, such as graphene-based emitters, because the emission response can be accurately measured independently of the light intensity. In the real-time communication measurements, the emitted light was detected by a normal-mode InGaAs APD photoreceiver (wavelength range: 1.0–1.6 µm, bandwidth: 1 GHz), in which the detected signal from the APD was amplified by a transimpedance amplifier and a pre-amplifier (100 MHz bandwidth), and was measured using a digital oscilloscope (see Supplementary Note 1). The eye pattern was also acquired using a digital oscilloscope under a random digital input signal. The frequency response in the real-time communication measurements are dominated by the bandwidth of a pre-amplifier for the light detection. For observation of emission images from arrayed emitters, the intensity of each emitter is tuned with a potentiometer in order to obtain the uniform intensity. For the emission measurement by the direct coupling of the capped light emitter to the multimode optical fiber, the optical fiber is manipulated to contact the emitter in air. Since all optical measurement are carried out by using InGaAs-based detectors (the CCD camera, the linear array detector, the Geiger-mode APD, and the normal-mode APD), their measurement results are suitably matched for the telecommunication wavelength band with fiber optics (1.26–1.63 µm).

**Data availability**. The data that support the findings of this study are available from the corresponding author upon request.

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

## Acknowledgements

The authors thank D. Tsuya, E. Watanabe and S. Tanigawa in NIMS, S. Honda, Y. Matsumoto, H. Ueno, H. Ishikuro, Y. Monnai and R. Mogi in Keio University, for technical support and discussions. This work was partially supported by PRESTO (Grant Number JPMJPR152B) and A-STEP from JST, KAKENHI (Grant Number 16H04355, 23686055, 26220604 and 16H00917) and Core-to-Core program from JSPS, SCOPE from MIC, the Cooperative Research Program of "Network Joint Research Center for Materials and Devices", Spintronics Research Network of Japan, and NIMS Nanofabrication Platform in Nanotechnology Platform Project by MEXT.

## Author contribution

Y.M., Y.A., T.Y and H.M. contributed to the experimental process. Y.M., Y.A., T.Y., R.R., K.I., and H.M. fabricated the device samples and carried out the measurements. H.A. and K.K. prepared a CVD-grown graphene. Y.M., Y.F. and H.M. carried out the simulations and theoretical interpretations. All authors have given approval to the final version of the manuscript.

## Additional information

**Competing interests:** The authors declare no competing interests.

