## [Peer Review File(PDF 638 kb) · Nature Communications]

Reviewers' comments:

Reviewer #1 (Remarks to the Author):

The authors report on the design, fabrication and characterisation of graphene based thermal emitters. They experimentally demonstrate very fast response times and use modelling, which includes heat transport by surface polar phonons, to simulate their results. Using a three layer device, they demonstrate real-time optical communication at 50 MHz. Whilst there have been previous studies of graphene based thermal emitters, there has not previously been studies showing, and analysing, such high response times. Although optical communication may not be the application for which these devices are most suited, I believe that the manuscript does make a significant contribution to the field and, with some amendments, is suitable for publication in Nature Communications. However, before publication, please can the authors address the following points:

- (1) In the title, and throughout the paper, the authors refer to graphene blackbody emitters. However, as the emissivity of graphene is only a few percent, it would perhaps be more accurate to refer to grey-body emitters. The authors should discuss the emissivity in the text.
- (2) Line 54. The authors state that graphene emitters have been demonstrated under steady-state or slow modulation. However, Mahlmeister et al (reference 25) demonstrated modulation of up to 100kHz, for a much larger device, which is still relatively fast for a thermal infrared emitter. Please can the authors therefore replace "slow modulation" with something like "relatively slow modulation (100kHz) in large area devices".
- (3) Line 78. Please can the authors state whether the DC bias voltage applied was continuous or pulsed.
- (4) Lines 141/142. Do the authors mean: "however, the existence of a hotspot by self is insufficient to explain the fast response of the experimentally observed emission"? Did the authors confirm that the position of the hotspot observed in the single layer graphene can be changed by the application of a gate bias?
- (5) Line 195. The authors attribute the slower temperature response of devices with multi-layer graphene to a higher value of g due to the "atomically-flat surface" of multi-layer graphene. Do the authors have experimental evidence that the multi-layer graphene sample is making better thermal contact with the substrate than single-layer graphene samples? Please could the authors state whether the increase in g in multi-layer samples is offset by an increase in the lateral thermal conductivity of the multi-layer graphene compared to single layer graphene? In addition, Mahlmeister et al (reference 25) showed that reducing the heat flow into the substrate using h-BN enabled the emission from larger area graphene thermal emitters to be modulated at higher frequencies. See comment (7) below.
- (6) Line 216. Please can the authors comment on whether there was any advantages in using 85 layer graphene? As stated in the SI, the emissivity is known to increase linearly with the number of layers, starting as approximately 2% for single layer graphene. In theory, there should be no further increase in emissivity after approximately 50 layers. Was the 85 layer graphene also grown by CVD, and how was the number of layers determined?
- (7) Paragraph beginning line 237. The authors discuss the use of Al₂O₃ to cap the emitters, stating that these emitters operate in air for more than 100 hours. However, Barnard et al [reference 24] described the use of hexagonal boron nitride as an encapsulation which allowed the operation in air of much larger area devices for more than 1000 hours. Please can the authors therefore expand this section to include a discussion of the possible failure mechanisms of their device, in particular the relationship between failure and device area.

Please can the authors state how many graphene layers there were in the encapsulated devices, and whether the deposition of the Al₂O₃ caused a change in the conductivity of the graphene.

Finally, Barnard et al encapsulated, both above and below, the graphene emitter with hexagonal boron nitride (h-BN). They showed that the h-BN is very effective in lateral heat transport, but

also that the h-BN underneath the graphene can be used to reduce the heat flow into the substrate. Mahlmeister et al (reference 25) showed that reducing the heat flow into the substrate using h-BN enabled the emission from larger area graphene thermal emitters to be modulated at higher frequencies. Please can the authors therefore add a comment on whether the use of different materials, including hexagonal boron nitride, would allow the design of larger area emitters that can be modulated at very high frequencies, making reference to the work of Barnard and Mahlmeister.

Reviewer #2 (Remarks to the Author):

Manuscript by Miyoshi et al. reports of high-speed on-silicon-chip graphene-based blackbody emitters for optical communications. The authors demonstrate high speed graphene light emitters at the near infrared spectrum, which overlaps telecommunication wavelengths. The NIR light from graphene-based devices is essentially from black-body like radiation from high temperatures achieved by joule heating. Their report is not first to report bright light emission from graphene-based devices, as the authors correctly cite previous works of others in the manuscript. They adequately support their claims with experiment and numerical calculations. What may be of interest to Nature Communication readership and beyond is their demonstration of their graphene-based light emitters at high switching speeds, as it opens up new avenues of application for graphene. In short, I recommend this manuscript to be accepted to be published as a regular article for Nature Communication if the authors respond and edit their manuscript in kind to my main concern:

As the authors state the importance of their work of being a graphene-based emitter for optical communications, I have some general technical issues on their experiment.

- (1) What bandwidth are counted by your photo-diode?
- (2) Has there been attempts to use a bandwidth filter to count/detect intensities corresponding to wavelengths/energies that is suitably matched for fiber optics?
- (3) The first two concerns is the arbitrary intensity plots the authors show. If low energy or long wavelength spectrum is counted by the photo-diode, then the authors claim for technological pathway and application for graphene-based emitters sit on a much weaker ground.

Reviewer #3 (Remarks to the Author):

The manuscript by Miyoshi and coworkers describes experimental results on implementing graphene-based high speed emitters. Optically broadband light emission is observed in the near-infrared wavelength regime. The light is emitted in the vertical direction and picked up using optical fibers. The authors demonstrate emitter arrays with up to 16 devices. Response times of 100 ps are reported, while data modulation with 50 MHz is shown, including an open eye diagram.

Thermal graphene emitters under electrical bias have been demonstrated in 2012 by Engel et al., Nat. Commun. 3, 906. This reference is missing, which I find quite surprising. Multiple emitters have also been shown before by the Hone group, ref. 23. The claimed novelty in the current manuscript is fast modulation with a rise time of 100 ps. Nevertheless, the actual modulation speeds are slow, in the MHz range. Also I fail to see how the measurement of an eye diagram can be considered real-time optical communication, nor how a 1Gbps modulation in arbitrary units supports the claim of high-speed modulation. As presented, I do not see any advantage in the demonstrate devices, neither in terms of speed, size or efficiency.

Therefore I do not think that the paper is suitable for Nature Communications and suggest referral to a technical journal. Before a resubmission elsewhere I would encourage the authors to address

a few further points

- 1) Can the authors further elucidate the modulation depth in Fig. 4c? Why do they not show the eye diagram at least at 1 Gbps? The authors should also measure the frequency response of the device and show the 3dB roll-off.
- 2) Pickup of the emitted light with a multimode fiber does not seem to be very practical. Instead the use of multiple single-mode fibers would be advantageous, such that individual optical channels can be discriminated.
- 3) Since the fast modulation is shown with an 85 layer device it would be interesting to show the time response of that device as well on a fast timescale. The curve in the SI is shown for a 1 μ s pulse duration, while the images in Fig.2d are on a 10 ns timescale.
- 4) The authors should think of a different acronym for surface polar phonons. SPP is already commonly in use for "surface plasmon polaritons".
- 5) Since silicon integration is a motivation of the current work, it is necessary to explain how the current devices are compatible with silicon photonics. Vertical emitters are not very practical for these applications.
- 6) When speaking about optical communication it is necessary to show some state-of-the-art modulation speeds. Since the authors claim potential 10 Gbps, it is necessary to either remove the claim or show the response at these speeds.
- 7) The authors should estimate the energy required for a single switching event. This determines in the end the power consumption of the device and indicates limitations for practical use.

Point-to-point response to the reviewers' comments:

Reviewer #1

The authors report on the design, fabrication and characterisation of graphene based thermal emitters. They experimentally demonstrate very fast response times and use modelling, which includes heat transport by surface polar phonons, to simulate their results. Using a three layer device, they demonstrate real-time optical communication at 50 MHz. Whilst there have been previous studies of graphene based thermal emitters, there has not previously been studies showing, and analysing, such high response times. Although optical communication may not be the application for which these devices are most suited, I believe that the manuscript does make a significant contribution to the field and, with some amendments, is suitable for publication in Nature Communications. However, before publication, please can the authors address the following points:

Response: We thank the referee for the high evaluation to our manuscript and suggestions for publication.

(1) In the title, and throughout the paper, the authors refer to graphene blackbody emitters. However, as the emissivity of graphene is only a few percent, it would perhaps be more accurate to refer to grey-body emitters. The authors should discuss the emissivity in the text.

Response: Accepted with thanks for the appropriate comment, and we revised the text as follows.

Revision: The following sentence is added in **Line 51**.

- **Line 51:** “Graphene-based blackbody emitters (grey-body emitters for thin graphene due to the low emissivity of 2.3 % per layer ¹⁷⁻²²) are also promising light emitters on silicon chip in NIR and mid-infrared region, just like CNT-based blackbody emitters.^{17-21,23-27}”

(2) Line 54. The authors state that graphene emitters have been demonstrated under steady-state or slow modulation. However, Mahlmeister et al (reference 25) demonstrated modulation of up to 100kHz, for a much larger device, which is still relatively fast for a thermal infrared emitter. Please can the authors therefore replace "slow modulation" with something like "relatively slow modulation (100kHz) in large area devices".

Response: Accepted, and the text is revised.

Revision: The following sentence and the **Ref. 26** (Ref. 25 in the original manuscript) are added in **Line 55**.

- **Line 55:** *“However, although graphene-based blackbody emitters have been demonstrated under steady-state conditions or relatively slow modulation (100 kHz) in large area devices²⁶, the transient properties of these emitters under high-speed modulation have not been reported to date.”*

(3) Line 78. Please can the authors state whether the DC bias voltage applied was continuous or pulsed.

Response: Accepted, and the text is revised.

Revision: The following word is added in **Line 81**.

- **Line 81:** *“Under continuous DC bias voltage”*

(4) Lines 141/142. Do the authors mean: "however, the existence of a hotspot by self is insufficient to explain the fast response of the experimentally observed emission"? Did the authors confirm that the position of the hotspot observed in the single layer graphene can be changed by the application of a gate bias?

Response: We agree with your comments, and the text is revised.

- On your first question, thank you for your appropriate comment. We have changed the description of the sentence in **Line 144**.

- On your second question, our devices are fabricated on “undoped” silicon substrate, as described in the “Method Device fabrication” section of main text (**Line 299**), because the substrate should be insulator for the design of coplanar waveguide with a characteristic impedance of 50Ω (as shown in **Line 299** and **Ref. 11**). Hence, we cannot apply a gate voltage in our high-speed graphene light emitters, unfortunately. However, we note that we fabricated the steady-state graphene light emitters, which were fabricated by the same fabrication method of this manuscript on a “doped” silicon substrate. (These devices cannot operate at high frequency because their electrodes are not coplanar electrodes. Instead, their electrodes are simple two-terminal electrodes whose characteristic impedance is not 50Ω .) As shown in the following figures, these emitters exhibit hot spot, and the hot spot can be moved by the application of a gate bias.

Fig. R1. NIR camera images of a graphene emitter at $V_g = -20, 0$ and 20 V.

Revision: Taking above discussion into account, we changed or added following sentences.

- **Line 144:** “However, the existence of a hotspot by itself is insufficient to explain the fast response of the experimentally observed emission (Figure 3b).”

- **Line 299:** “These electrodes were designed to be coplanar waveguide with a characteristic impedance of 50Ω on an undoped silicon substrate for high-speed light emitters (except for the array device in Figure 5a and 5b).¹¹ A gate bias cannot be applied because of the use of an undoped substrate.”

(5) Line 195. The authors attribute the slower temperature response of devices with multi-layer graphene to a higher value of g due to the “atomically-flat surface” of multi-layer graphene. Do the authors have experimental evidence that the multi-

layer graphene sample is making better thermal contact with the substrate than single-layer graphene samples? Please could the authors state whether the increase in g in multi-layer samples is offset by an increase in the lateral thermal conductivity of the multi-layer graphene compared to single layer graphene? In addition, Mahlmeister et al (reference 25) showed that reducing the heat flow into the substrate using h-BN enabled the emission from larger area graphene thermal emitters to be modulated at higher frequencies. See comment (7) below.

Response: Thank you very much for your instructive comment. It is interesting question whether the increase in g can be offset by an increase in the lateral thermal conductivity (κ) of multi-layer graphene (i.e., the increase in κ can be equated with the increase in “ g ” for multi-layer graphene). We considered your question, and our conclusion is that the increase of κ cannot be explained by the increase in g because the effect of κ increase in “transient” temperature responses is different from that of g increase. When g is increased, the graphene temperature is strongly affected by the substrate temperature due to strong thermal coupling between graphene and substrate. This causes the emission response change of the second slow region in Figure 3d (indicated by blue arrows). As shown in the figure, the second slow region is emphasized in the time-resolved emission result (also described in **Line 169**) for a high g . On the other hand, when κ is increased, the lateral thermal conduction has a very small effect on the emission response of the second slow region in Fig. 3d because the lateral thermal conduction does not contribute to the increase of the substrate temperature. Hence, the increase in g and κ can be distinguished by the transient temperature response and the time-resolved emission measurement.

Taking the above discussions into account, we can also answer following reviewers’ question.

Do the authors have experimental evidence that the multi-layer graphene sample is making better thermal contact with the substrate than single-layer graphene samples?

Our measurement results in Figure 2d, which is time-resolved emission for single-, few-, and multi-layers graphene, indicates that the second slow response is enhanced for multi-layer graphene. This is direct evidence that g (i.e., thermal conductance between graphene and substrate) of the multi-layer graphene is higher than that of the single-layer graphene.

These results are consistent with the previous report (in **Ref. 40**, by K. Mak, C. Lui and T. Heinz) on the direct thermal conductance measurements of the graphene/SiO₂ interface, where single graphene exhibits lower thermal conductance than multi-layer graphene.

(Just for information, we note that the reviewers' comments are partially right as follows.

- If only the “steady-state” temperature is taken into account, it is difficult to distinguish the effect of increase of g and κ because the increase in g and κ decrease the “steady-state” temperature of graphene.
- If we focus on the “initial fast response” in Fig. 3d (indicated by green arrows), this initial fast response speed can be increased by the increase in g and κ . Hence, it is difficult to distinguish the effect of g and κ in the “initial fast response time” of the emission.)

The response to the following reviewers' question on “h-BN”

In addition, Mahlmeister et al (reference 25) showed that reducing the heat flow into the substrate using h-BN enabled the emission from larger area graphene thermal emitters to be modulated at higher frequencies. See comment (7) below.

will be described later.

Revision: Above discussion indicates that the time-resolved emission measurement (i.e., transient temperature response measurement) is an effective method to investigate the in-plane and out-of-plane thermal transport (shown in **Line 210**) in the graphene device. Using our method, we could directly elucidate the effect of the thermal contact g in the graphene emitters. Taking this into account, the text is revised as follows.

- **Line 199:** *“We note that not only g but also κ of multi-layer graphene is higher than that of single-layer graphene;¹⁶ however, the distorted emission response of multi-layer graphene in Figure 2d cannot be explained by the model without high g .”*

- **Line 210:** *“Our results imply that the direct measurement of the transient temperature is an effective method to precisely elucidate the in-plane and out-of-plane thermal transport in the graphene devices, and the time-resolved emission measurement, demonstrated in this study, is one of the effective method for the transient temperature investigation.”*

(6) Line 216. Please can the authors comment on whether there was any advantages in using 85 layer graphene? As stated in the SI, the emissivity is known to increase linearly with the number of layers, starting as approximately 2% for single layer graphene. In theory, there should be no further increase in emissivity after approximately 50 layers. Was the 85 layer graphene also grown by CVD, and how was the number of layers determined?

Response: As shown in **Line 289** in the “Method” section, 85 layer graphene is prepared by the mechanical exfoliation method with scotch tape, which is very difficult to precisely control the number of graphene layers. The layer number of prepared graphene is measured by atomic force microscope (in **Line 291**). In the simple theoretical model, the emission intensity linearly increase with increasing the number of layers in the range of a few tens layers, and the intensity gradually saturate in the range of 50 to 100 layers, as commented by reviewer and shown in **Ref. 25 and 50**. We used 85 layer graphene, which is chosen as the criterion of “less than 100 layers”.

Revision: We added two references (**Ref. 25 and 50**) in the main text.

- Line 223: *“In this measurement, multi-layer graphene (~ 85 layers) was used because of higher emission intensity and higher durability of the graphene devices under emission with increasing the number of graphene layers.^{26,51}”*

(7) Paragraph beginning line 237. The authors discuss the use of Al₂O₃ to cap the emitters, stating that these emitters operate in air for more than 100 hours. However, Barnard et al [reference 24] described the use of hexagonal boron nitride as an encapsulation which allowed the operation in air of much larger area devices for more than 1000 hours. Please can the authors therefore expand this section to include a discussion of the possible failure mechanisms of their device, in particular the relationship between failure and device area.

Response: We agree with reviewers’ comments.

As commented by reviewer, we didn’t cite the important report of **Ref. 21** (reference 24 in the original manuscript), which is study on the long lifetime operation for more than

1000 hours for large area devices. Therefore, we added this reference and a discussion in the paragraph on Al₂O₃-capped emitters (beginning **Line 246**). The failure mechanism of graphene emitters is mainly the damage of graphene by reaction with oxygen in air. The capping layer on graphene is effective to avoid this reaction. The advantage of Al₂O₃ capping by atomic layer deposition (ALD) is that this ALD method is conventional semiconductor process, which is used in the silicon-based integrated circuit technology. In our study, we have not carried out the device-area dependence of the lifetime; therefore, it is not clear the relationship between failure and device area, and further study is needed to clear this.

Revision: We added **Ref. 21** and the text is revised in the paragraph on Al₂O₃ cap (beginning **Line 246**) as follows.

- **Line 246:** *“Although all graphene emitters described above are measured in a high vacuum to avoid the damage of graphene by reaction with oxygen in air, we fabricated the graphene emitters encapsulated by Al₂O₃ insulator. Such encapsulation technique was previously demonstrated by using hexagonal boron nitride (h-BN), and this technique allowed the operation of large area devices for more than 1000 hours in air.²¹ We grew the Al₂O₃ capping layer by an atomic layer deposition (ALD) method, which is used in a conventional silicon-based semiconductor process. The conductance of the device was decreased to ~ 50 % by the deposition of Al₂O₃ due to charged impurity scattering and SPoPh scattering by the capping layer.⁵³ As shown in Figure 5c, the Al₂O₃-capped emitters can operate in air for more than 100 hours. Furthermore, by using this Al₂O₃-capped emitters, we also demonstrated the direct coupling of this capped light emitter to a multimode optical fiber in air owing to their small footprint and planar device structure (Figure 5d). High-speed light emission could be directly observed through this optical setup (Figure 5e).”*

Please can the authors state how many graphene layers there were in the encapsulated devices, and whether the deposition of the Al₂O₃ caused a change in the conductivity of the graphene.

Response: In the encapsulated devices, three-layer graphene was used. In this device, the conductance of the device was decreased to ~ 50 % by the deposition of the Al₂O₃ due to

charged impurity scattering and SPoPh scattering by the capping layer as reported in **Ref. 53**.

Revision: We added the description of number of graphene layers in the caption of Figure 5c. The **Ref. 53** and the description on the conductivity change by the deposition of the Al_2O_3 are added in the text as follows.

- **Line 252:** *“The conductance of the device was decreased to ~ 50 % by the deposition of Al_2O_3 due to charged impurity scattering and SPoPh scattering by the capping layer.”⁵³”*

- **Line 568 in the caption of Figure 5c:** *“c, NIR camera images of the light emission from Al_2O_3 -capped emitters with three-layer graphene after 0 and 100 hours of continuous work in air.”*

Finally, Barnard et al encapsulated, both above and below, the graphene emitter with hexagonal boron nitride (h-BN). They showed that the h-BN is very effective in lateral heat transport, but also that the h-BN underneath the graphene can be used to reduce the heat flow into the substrate. Mahlmeister et al (reference 25) showed that reducing the heat flow into the substrate using h-BN enabled the emission from larger area graphene thermal emitters to be modulated at higher frequencies. Please can the authors therefore add a comment on whether the use of different materials, including hexagonal boron nitride, would allow the design of larger area emitters that can be modulated at very high frequencies, making reference to the work of Barnard and Mahlmeister.

Response: Thank you for your instructive comments. We agree with your comments and have reconsidered the “summary” paragraph (**Line 272**). As a result, the quality of our manuscript could be improved due to your advice.

In our manuscript, we mainly focus on the control of the emission properties by changing the number of graphene layers. On the other hand, as commented by reviewer, the thermal transport control by the modification of the “substrate” is also promising method to control the emission properties. For example, since the thermal transport through surface polar phonon (SPoPh) can happen for the polar substrate such as not only SiO_2 but also HfO_2 , Al_2O_3 and h-BN (**Ref. 33, 54**), the effect of SPoPh can be controlled by changing the substrate. The physical contact (i.e., flatness of graphene) is also different

depending on the substrate materials (**Ref. 49, 55**). For example, as mentioned by reviewer, graphene on two-dimensional h-BN substrate exhibits unique thermal properties such as flat contact with graphene, anisotropic thermal properties, two-dimensional phonon transport, compared with SiO₂ substrate. Especially, the unique properties of polaritons in two-dimensional materials have recently attracted attention as reported in **Ref. 56**. The unique thermal properties of low-dimensional substrate such as h-BN are applicable to the light emitters as reported in **Ref. 21 and 26** (reference 25 in the original manuscript).

Revision: Taking above discussion into account, we added the references and revised the text in the “summary” paragraph as follows.

- Line 272: *“Since SPoPhs can be excited in the polar substrate such as HfO₂, Al₂O₃ and h-BN,^{33,54} the effect of SPoPh can be modified by changing the substrate. In particular, two-dimensional materials such as h-BN have recently attracted attention owing to their unique properties of their flat surfaces,^{49,55} anisotropic structure and two-dimensional polariton transport,⁵⁶ and these unique properties are applicable to the graphene light emitters.^{21,26}”*

Reviewer #2

Manuscript by Miyoshi et al. reports of high-speed on-silicon-chip graphene-based blackbody emitters for optical communications. The authors demonstrate high speed graphene light emitters at the near infrared spectrum, which overlaps telecommunication wavelengths. The NIR light from graphene-based devices is essentially from black-body like radiation from high temperatures achieved by joule heating. Their report is not first to report bright light emission from graphene-based devices, as the authors correctly cites previous works of others in the manuscript. They adequately support their claims with experiment and numerical calculations. What may be of interest to Nature Communication readership and beyond is their demonstration of their graphene-based light emitters at high switching speeds, as it opens up new avenues of application for graphene. In short, I recommend this manuscript to be accepted to be published as a regular article for Nature Communication if the authors respond and edit their manuscript in kind to my main concern:

Response: We thank the reviewer for the high evaluation to our manuscript and recommendation it to be accepted to be published.

As the authors state the importance of their work of being a graphene-base emitter for optical communications, I have some general technical issues on their experiment.

(1) What bandwidth are counted by your photo-diode?

Response: Spectral response range (wavelength bandwidth) of normal-mode InGaAs avalanche photodiode for the real-time communication is 1.0 to 1.6 μm .

Revision: We added the wavelength bandwidth in the “Method” paragraph as follows. (We note that we also added the “frequency bandwidth (i.e., detection speed)” of the photoreceiver for the response to Reviewer 3.)

- Line 324: *“In the real-time communication measurements, the emitted light was detected by a normal-mode InGaAs APD photoreceiver (wavelength range: 1.0 to 1.6 μm , bandwidth: 1 GHz), in which the detected signal from the APD was amplified by a*

transimpedance amplifier (TIA) and a pre-amplifier (100 MHz bandwidth), and was measured using a digital oscilloscope (Figure S1b)."

(2) Has there been attempts to use a bandwidth filter to count/detect intensities corresponding to wavelengths/energies that is suitably matched for fiber optics?

Response: We thanks for the reviewers' comment. The typical telecommunication wavelength band with optical fibers ranges from 1.26 to 1.63 μm . This wavelength bandwidth is almost the same bandwidth as our InGaAs detectors as described in the "Method" paragraph. Hence, all optical measurement results in our manuscript are suitably matched for the telecommunication band with fiber optics.

Revision: We revised the text in the "Method" paragraph as follows.

- **Line 334:** *"Since all optical measurement are carried out by using InGaAs-based detectors (the CCD camera, the linear array detector, the Geiger-mode APD and the normal-mode APD), their measurement results are suitably matched for the telecommunication wavelength band with fiber optics (1.26 to 1.63 μm)."*

(3) The first two concerns is the arbitrary intensity plots the authors show. If low energy or long wavelength spectrum is counted by the photo-diode, then the authors claim for technological pathway and application for graphene-based emitters sit on a much weaker ground.

Response: We thanks for the reviewers' comment. As mentioned above, all measurement results in our manuscript are suitably matched for the telecommunication wavelength band with fiber optics. The detection efficiency of the InGaAs detector is almost zero in the wavelength range over 1.7 μm .

Revision: We added the above two sentences in the "Method" paragraph (**Line 324, 334**).

Reviewer #3 (Remarks to the Author):

The manuscript by Miyoshi and coworkers describes experimental results on implementing graphene-based high speed emitters. Optically broadband light emission is observed in the near-infrared wavelength regime. The light is emitted in the vertical direction and picked up using optical fibers. The authors demonstrate emitter arrays with up to 16 devices. Response times of 100 ps are reported, while data modulation with 50 MHz is shown, including an open eye diagram.

Thermal graphene emitters under electrical bias have been demonstrated in 2012 by Engel et al., Nat. Commun. 3, 906. This reference is missing, which I find quite surprising.

Response: This reference is added in accordance with reviewers' suggestion.

(Note that this paper is study on “steady-state” light *detection* and *radiation* in graphene *transistors* with “micro-cavity” at short wavelength region (i.e., *not telecommunication wavelength*); therefore, this paper has basically little relevance to our manuscript, which is study on the “high-speed” modulation and “optical communication” with graphene *emitters at telecommunication wavelength*.)

Revision: The following reference is added in the text of **Line 53 and 407**.

- **Line 407:** “27. Engel, M., Steiner, M., Lombardo, A., Ferrari, A. C., Löhneysen, H., Avouris, P. & Krupke, R. *Light–matter interaction in a microcavity-controlled graphene transistor. Nat. Commun. 3, 906 (2012).*”

Multiple emitters have also been shown before by the Hone group, ref. 23.

Response: Thank you for your suggestion. The description of the novelty of our array devices was insufficient in our original manuscript. Therefore, we improved the text in order to show clearly the following novelty.

The multiple emitters reported by Hone group in **Ref. 20** (Ref. 23 in the original manuscript) was “one-dimensional” linear array of graphene emitters. On the other hand, our multiple emitters are “two-dimensional” array of graphene emitters. In addition, our

array emitters have “higher density” than in the previous report of **Ref. 20** by using “unsuspended” graphene.

Revision: We revised the text in order to clearly show the novelty of our array devices.

- **Line 237:** *“We fabricated two-dimensional array of graphene emitters, which are high dimension and density by using unsuspended graphene compared with earlier work.”²⁰”*

The claimed novelty in the current manuscript is fast modulation with a rise time of 100 ps. Nevertheless, the actual modulation speeds are slow, in the MHz range.

Also I fail to see how the measurement of an eye diagram can be considered real-time optical communication, nor how a 1Gbps modulation in arbitrary units supports the claim of high-speed modulation. As presented, I do not see any advantage in the demonstrate devices, neither in terms of speed, size or efficiency.

Therefore I do not think that the paper is suitable for Nature Communications and suggest referral to a technical journal.

Response: We disagree on this reviewers’ comment. The reviewer misunderstand the important results and novelty of our study.

The biggest misunderstanding is that the response speeds of the graphene emitters strongly depend on the “number of layers” of graphene, as indicated in Figure 2d and 3d,e and discussed in the paragraph beginning **Line 104, 155 and 191**; that is, the emitter with GHz and MHz speeds are different samples. As described in these results and paragraph, the graphene emitters with “single- and few-layer graphene” exhibit high-speed emission with the response time of 100 ps, because the emission response is dominated by the initial fast region of the emission response (indicated by the green arrows of Fig. 2d). On the other hand, the graphene emitters with “multi-layer graphene” exhibit the slower response in the MHz range because the emission response is dominated by the second slow region of the emission (indicated by the blue arrows of Fig. 2d). These results indicates that the response speed is different from the single- (few-) layer graphene device and multi-layer graphene device; however, the reviewer confused these fast and slow samples.

In addition, the reviewer misunderstand the measurement method of the “actual modulation speeds”. The “actual modulation speeds” of the light emitters can be fundamentally measured not by the “real-time optical communication measurement (shown in Fig. 4a, b)” but by “the time-resolved emission measurement based on a single

photon counting method (shown in Fig. 2, **Line 90 and 317**)". As will be described below, the response speeds measured by "real-time measurement" become slower than the "actual modulation speeds" for the low-intensity light sources such as micro light sources, and these speeds obtained from "real-time measurement" are dominated not by the response speed of emitters but by that of "detectors". In fact, the response of Figure 4a, which is the real-time communication result at 50 MHz for three-layer-graphene device, is dominated not by the response speed of the graphene emitter but by the bandwidth of a "pre-amplifier in a photoreceiver" (100 MHz). (In an optical communication theory, the communication speed should be decreased with decreasing the light intensity as the light intensity approaches the "*Noise Equivalent Power (NEP) of a photodetector*", because the white noise of the detector, which has a flat spectrum over the whole frequency range, should be reduced by low pass filter under the condition of the low light intensity.) On the other hand, in order to measure the "actual modulation speeds" of micro light sources (e.g., emission from single quantum dot of semiconductor, nanomaterial such as nanocarbon materials and molecule), the "photon counting method" is fundamentally used because the "actual" response speed of the low-intensity light sources can be accurately measured independently of the light intensity. In fact, we could measure the accurate response speed by the "photon counting method" as shown in Figure 2, and these results indicates that the "actual" high-speed emission with the response time of 100 ps can be realized by using single- and few-layer graphene emitters.

Finally, we would like to emphasize that the modulation speeds not only in the GHz range but also in the MHz range is the fastest speed of the graphene emitters compared with the previous reports, where the fastest speed is "100 kHz". (Note that this 100 kHz modulation is not the real-time optical communications.) The GHz-modulations of our "blackbody" emitters are more than 10^4 and 10^7 times faster than the graphene-based blackbody emitters and the conventional blackbody emitters with metal filaments, respectively. In addition, we also demonstrated the "real-time optical communications" at 1 and 50 MHz, which are the first report of "real-time communication measurement" with graphene emitters. Although the real-time measurement is difficult measurement technique, the speeds of our "real-time" communication at 1 and 50 MHz are 10 and 500 times faster than the previous modulation of 100 kHz, which is not real-time measurement, respectively. Furthermore, we theoretically found the novel mechanism of the fast thermal response, which is dominated by the quantum and remote heat transfer through SPoPh. This is the first report on the fast heat transport mechanism in graphene devices.

Taking above discussion into account, we disagree with the reviewers' comments of *"As presented, I do not see any advantage in the demonstrate devices, neither in terms of speed, size or efficiency."*

Revision: We revised the text as follows in order to show clearly (i) the graphene layer-number dependence of the emitters' response speed, (ii) the difference between "real-time measurement" and "photon counting measurement" and (iii) the novelty of high-speed emission of our graphene emitters.

- **Line 27:** *"We also experimentally demonstrated first real-time optical communications at 1 and 50 Mbps with multi- and few-layer graphene, respectively"*

- **Line 60:** *"Here, we report the first study on a highly integrated, high-speed and on-chip blackbody emitter based on graphene in NIR region including telecommunication wavelength."*

- **Line 71:** *"Moreover, we experimentally demonstrate first optical communications at 1 and 50 Mbps based on eye-pattern analysis and real-time waveform detection with multi- and few-layer graphene, respectively"*

- **Line 317:** *"In the time-resolved emission measurements based on a single photon counting method,"*

- **Line 321:** *"The time-resolved measurements based on a single photon counting method is effective method to investigate the response speed of the low-intensity light sources, such as graphene-based emitters, because the emission response can be accurately measured independently of the light intensity."*

Therefore I do not think that the paper is suitable for Nature Communications and suggest referral to a technical journal. Before a resubmission elsewhere I would encourage the authors to address a few further points.

Response: We disagree with this reviewers' comment as mentioned above because the reviewers' suggestion is obtained based on the misunderstood results, measurement methods and novelty of our study.

1) Can the authors further elucidate the modulation depth in Fig. 4c? Why do they not show the eye diagram at least at 1 Gbps? The authors should also measure the frequency response of the device and show the 3dB roll-off.

Response: Thank you for reviewers' comment. Since our explanation of Fig. 4c in the main text and the figure caption of the original manuscript was insufficient, the results of Fig. 4b and 4c were confusing; that is, the samples (the number of graphene layers) and the measurement methods are different between Fig. 4b and 4c. Therefore, we added the explanation of Fig. 4c, as mentioned below.

As shown in Fig. 4c and this caption, the result in Fig. 4c is measured by a "photon counting method", which is not the real-time communication as discussed above. In the a photon counting method, the measured emission response curves are given by the histograms of the photon counts, and the numbered label of ordinate axis should be normalized as shown in Fig. 2, 3b, 4c and 5e. Therefore, the unit of the measurable physical quantity (such as "voltage" in Fig. 4a,b) cannot be used as the ordinate label of the photon counting result; therefore, the modulation depth of the physical quantity, which is suggested by reviewer, cannot be shown in Fig. 4c, unfortunately.

Concerning on the second question that "*Why do they not show the eye diagram at least at 1 Gbps?*", the "eye diagram" cannot be measured at 1 Gbps because the multi-layer graphene, which has a high emission intensity but a slow response speed (\sim MHz), should be used for the eye diagram measurement. The eye diagram cannot be measured by using single- and few-layer graphene, which has a low emission intensity and high response speed (\sim GHz), because the light intensity is insufficient for the eye diagram measurement. It is important that the "eye diagram" measurement is one of the "real-time" communication measurements but is the hardest difficulty measurement in the real-time communication measurements. Since the high-intensity light source is required for the "eye diagram" measurement, the "eye diagram" measurement is extremely difficult measurement technique compared with the "photon counting" measurement.

For the emitters with three-layer graphene, we demonstrated the fast modulation with the response time of 100 ps (shown in Fig. 2 and S3) and the continuous modulation under a 1 Gbps input (shown in Fig. 4c) by the photon counting measurement; however, the real-time measurement is demonstrated at "50 MHz" as shown in Fig. 4a. This slow modulation of the real-time measurement compared with the photon counting measurement is due to the use of a high-gain, low-noise and narrow bandwidth (100 MHz)

pre-amplifier for the measurement of the low-intensity emitters with thin graphene; that is, the speed of the “real-time communication” strongly depend on the emission intensity (as discussed above). The eye-diagram measurement at 1 Gbps, which is suggested by reviewer, is technically impossible for single- and three-layer graphene because of the insufficient emission intensity for eye-diagram measurement.

On the other hand, the emitter with 85-layer graphene can exhibit both photon-counting (Fig. S4) and eye-diagram measurements (Fig. 4b) at 1 MHz because the emitters with thick graphene can emit high-intensity light. Taking these results into account, we described the demonstrations of “high-speed modulation at 100 ps” and “real-time optical communication at 1 and 50 Mbps” in the abstract, main text, and summary of our manuscript. To show clearly the difference of the number of graphene layers in the photon counting and real-time measurements, we added the information of number of graphene layers in the abstract, main text, and summary of our manuscript.

On the third question of “3dB roll-off”, the direct measurement of the bandwidth of the graphene emitter with a spectrum analyzer is impossible for high-speed emitters, because the emission intensity is insufficient for the direct measurement of bandwidth with a real-time photoreceiver and a spectrum analyzer in the GHz range. However, it is well known that the 3 dB roll-off frequency can be theoretically calculated from the rise time t_r of the emitter, which is given by $\sim 0.35/t_r$. The roll-off frequency (i.e., bandwidth) of the emitter is estimated to be ~ 3 GHz by using the rise time of 100 ps.

Revision: We revised the main text and the figure caption of Fig. 4 in order to show clearly the measurement methods and the details of measured samples. We revised the brief title sentence in the beginning of Fig. 4, the figure caption of Fig. 4c and the main text to avoid the confusion of the measurement method of Fig. 4c. We also added the information of number of graphene layers in the abstract, main text, and summary of our manuscript to avoid the confusion of the sample difference in the photon counting and real-time measurements. In addition, we added the bandwidth of the light detection, which dominates the frequency response of the real-time communications in the “Method” paragraph.

- **Line 553:** “Figure 4. Real-time optical communications and continuous modulation.”

- **Line 557:** “c, High-speed emission modulation (red curve) from three layer graphene device under a continuous input (0 V - 6 V in height) of 1 Gbps (green curve) by time-resolved emission measurements.”

- **Line 27:** “We also experimentally demonstrate first real-time optical communications at 1 and 50 Mbps with multi- and few-layer graphene, respectively”

- **Line 71:** “Moreover, we experimentally demonstrate first optical communications at 1 and 50 Mbps based on eye-pattern analysis and real-time waveform detection by using multi- and few-layer graphene, respectively”

- **Line 264:** “In addition, we have reported the first application of graphene-based emitters for optical communications via optical fiber with the real-time signal transmission (1 and 50 Mbps) by using multi- and few-layer graphene, respectively.”

- **Line 324:** “In the real-time communication measurements, the emitted light was detected by a normal-mode InGaAs APD photoreceiver (wavelength range: 1.0 to 1.6 μm , bandwidth: 1 GHz), in which the detected signal from the APD was amplified by a transimpedance amplifier (TIA) and a pre-amplifier (100 MHz bandwidth), and was measured using a digital oscilloscope (Figure S1b).”

- **Line 329:** “The frequency response in the real-time communication measurements are dominated by the bandwidth of a pre-amplifier for the light detection.”

2) Pickup of the emitted light with a multimode fiber does not seem to be very practical. Instead the use of multiple single-mode fibers would be advantageous, such that individual optical channels can be discriminated.

Response: We disagree on this reviewers' comment. The reviewer misunderstand the fundamental properties of multimode fibers and our device structure and properties. The reviewers' suggestions of “the use of multiple single-mode fibers” does not make sense and are technically impossible as follows. To avoid the confusion between “step-index” and “graded-index” multimode fibers, we added the word of “graded-index” multimode fibers, which is used in our study, in the “Method” section.

The multimode fibers (MMFs), which means “graded-index” multimode fiber in the field of “optical communications”, support a range of transmission lengths of > 300 m at 10 Gbps communication. In our measurement, the lengths of MMFs are ~ 1m, which is adequately satisfy the supported transmission length of MMFs; therefore, there is no problem with the use of MMFs in our time-resolved and real-time emission measurement. We cannot understand the reviewers’ comment that “*Pickup of the emitted light with a multimode fiber does not seem to be very practical.*”, because MMFs are practically used in the optical communication over short distances, such as within a campus, building, rack, board. Please see the details of multimode fibers below.

http://cdn.intechopen.com/pdfs/45023/InTech-Multimode_graded_index_optical_fibers_for_next_generation_broadband_access.pdf

On the reviewers’ suggestion on “multiple single-mode fibers (multicore fibers?)”, this suggestion is pointless because most of our results are for the single channel devices as shown in Fig. 1, 2, 3, 4 and 5c,d,e. In addition, for our array emitters shown in Fig. 5a,b, the reviewers’ suggestion that “*Instead the use of multiple single-mode fibers would be advantageous, such that individual optical channels can be discriminated.*” is technically impossible because the “pitch” of our emitter array of 3 μm (shown in Fig. 5a,b) is extremely narrower than not only the pitch of the conventional multiple single-mode fibers (multicore fibers) of ~ several tens μm but also the core diameter of ~ 10 μm . This is the important advantage of our array emitters in our study, because our array devices can be applied to the highly-integrated optoelectronic devices such as silicon photonics.

Revision: To avoid the confusion between “step-index” and “graded-index” multimode fibers, we added the word of “graded-index” multimode fibers in the “Method” section.

- **Line 320:** “*a multimode graded-index optical fiber (MMF)*”

3) Since the fast modulation is shown with an 85 layer device it would be interesting to show the time response of that device as well on a fast timescale. The curve in the SI is shown for a 1 us pulse duration, while the images in Fig.2d are on a 10 ns timescale.

Response: The reviewers' suggestions of "showing the time response of the 85 layer device as well on a fast timescale" does not make sense because the time-resolved emission for "multi-layer" graphene is already shown in Fig. 2d (indicated by "28 layers") and discussed in the main text.

Although we added the time-resolved emission result for 85-layer device on a 5 ns timescale in Supplementary Information in accordance with the reviewers' comment, there is no significant difference between the results in Fig. 2d and Fig. S4b. The mechanisms of the slow response emission for multi-layer graphene is describe in the text. (In short, the time-resolved emission response is dominated by the second slow response indicated by blue arrows in Fig. 2d.)

Revision: In accordance with the reviewers' comment, we added the time-resolved emission result in Fig. S4b in Supplementary Information. (Note that this result is no significant difference from "28 layers" in Fig. 2d).

Figure S4 Time-resolved emission under rectangular input of (a) 1 μs and (b) 5 ns in width and 0 - 2.4 V in height for the emitter with 85-layer graphene, which is used for an eye-pattern measurement in Figure 4b. The result of a can explain the behavior of the observed eye pattern. The result of b shows no significant difference from the result of "28 layers" in Fig. 2d, which indicates the time-resolved emissions for "multi-layer" graphene device.

4) The authors should think of a different acronym for surface polar phonons. SPP is already commonly in use for "surface plasmon polaritons".

Response: Thank you for reviewers' comment. In accordance with the reviewers' comment, the acronym for surface polar phonons is changed from SPP to SPoPh. (Note that SPP is also used for "surface polar phonon" in the research field of semiconductor devices, e.g., **Ref. 33**)

Revision: The acronym for surface polar phonons is changed from SPP to SPoPh in our manuscript.

5) Since silicon integration is a motivation of the current work, it is necessary to explain how the current devices are compatible with silicon photonics. Vertical emitters are not very practical for these applications.

Response: Thank you for reviewers' comment. Nanocarbon-based light emitters are advantageous over the compound-semiconductor emitters for high-density optoelectronic devices integrated with silicon-based platform because (i) micro-light emitters can be directly integrated on silicon chips with the narrow pitch of $\sim 1\mu\text{m}$, which is $< 1/10$ times narrower than semiconductor light emitter array, (ii) the graphene emitters have very thin and planar device structure, where the emitting layer of the graphene is exposed, in contrast to the semiconductor light emitters with the emitting layer embedded in a p-n junction. These advantages enable the graphene emitters to be combined with the silicon-based integrated platform such as "silicon photonics" and "monolithic three-dimensional optoelectronic integrated circuit".

For "silicon photonics", the direct, near-field coupling of the emitted light from graphene into a waveguide can be realized, as opposed to the conventional far-field fiber coupling of an external semiconductor light source, because the emitting layer of graphene can be directly contact on the top surface of a silicon waveguide due to their small footprint and exposed structure of the graphene emitters. This can open new routes to highly integrated silicon photonics. This technique of direct coupling between a nanocarbon material and a silicon waveguide was demonstrated in carbon nanotube light emitters reported in **Ref. 2, 8** and in graphene photodetectors reported in **Ref. 5**.

For "monolithic three-dimensional optoelectronic integrated circuit", the "vertical" emission from very thin graphene emitters can be used for inter-layer optical communication (interconnection) in the silicon-based 3D integrated circuit, which is a n integrated circuit manufactured by stacking silicon wafers and interconnecting them

vertically as a next-generation silicon technology. This technique with “vertical emitters” was demonstrated in carbon nanotube light emitters reported in **Ref. 4**.

These indicate that the graphene emitters integrated on silicon chips are compatible with silicon-based photonics and optoelectronics such as silicon photonics and 3D integrated circuits.

Revision: In accordance with reviewers’ comments and above discussion, we added the description that the nanocarbon optoelectronic device including the graphene emitters are compatible with silicon-based photonics and optoelectronics as follows.

- **Line 279:** *“In addition, it has been reported that the optoelectronic devices with nanocarbon materials are compatible with integrated silicon photonics and three-dimensional integrated circuits.^{2,4,5,8} Hence, graphene-based emitters can be directly combined with the silicon platform with three-dimensional integrated electronics and integrated silicon photonics, and they can open new routes to highly integrated optoelectronics, e.g. optical interconnects.”*

6) When speaking about optical communication it is necessary to show some state-of-the-art modulation speeds. Since the authors claim potential 10 Gbps, it is necessary to either remove the claim or show the response at these speeds.

Response: Although the response time of 100 ps corresponds to the frequency response of 10 GHz as reported in **Ref. 12**, we removed the description of 10 GHz in accordance with the reviewers’ comment. The 100-ps response time of our graphene emitters is fast and comparable in comparison to the conventional semiconductor light emitting diodes (~ MHz) and laser diodes (~ GHz), respectively. Our graphene emitters exhibit the fastest speed compared to state-of-the-art graphene emitters (~ 100 kHz). (Note that the “real-time” communication speed (~ MHz) with the graphene emitter is dominated by the bandwidth of pre-amplifier in the light detection circuit as discussed above.)

Revision: The descriptions of “10 GHz” are removed or are replaced by “100-ps response time” from our manuscript. We added the description of response time of our graphene emitters compared with the semiconductor light sources.

- **Line 260:** *“In summary, we have demonstrated graphene-based blackbody emitters, which have a high modulation speed (100-ps response time) and small footprint ($\sim 1 \mu\text{m}^2$) at NIR wavelength including telecommunication wavelength.”*

- **Line 262:** *“The 100-ps response time of this graphene emitter is fast and comparable in comparison to the conventional semiconductor light emitting diodes (\sim MHz) and laser diodes (\sim GHz), respectively.”*

7) The authors should estimate the energy required for a single switching event. This determines in the end the power consumption of the device and indicates limitations for practical use.

Response: Interestingly, little energy is required for a single switching event for graphene emitters because the graphene emitter is the “resistive” device with extremely small electrical capacitance, as opposed to the large capacitance in a p–n junction for a semiconductor light emitting diode (LED). For LEDs, the energy is required for a single switching event because of capacitive charge and discharge in a p-n junction. On the other hand, the graphene device can emit without a p-n junction; therefore, the graphene emitter exhibits no energy loss of charge and discharge for a switching event because of its little electrical capacitance.

REVIEWERS' COMMENTS:

Reviewer #1 (Remarks to the Author):

I believe that the authors have addressed the comments and issues raised by the referees, and recommend that the manuscript is published within Nature Communications.

Reviewer #2 (Remarks to the Author):

Authors satisfactorily addressed my concerns in reply and in revising their manuscript.